# A Robotic Experimental Setup with a Stewart Platform to Emulate Underwater Vehicle-Manipulator Systems

**DOI:** 10.3390/s22155827

**Published:** 2022-08-04

**Authors:** Kamil Cetin, Harun Tugal, Yvan Petillot, Matthew Dunnigan, Leonard Newbrook, Mustafa Suphi Erden

**Affiliations:** 1Institute of Sensors, Signals and Systems, School of Engineering and Physical Sciences, Heriot-Watt University, Edinburgh EH14 4AL, UK; 2Edinburgh Centre for Robotics, Edinburgh EH14 4AL, UK

**Keywords:** underwater vehicle–manipulator system, robotics emulator, contact management, remote inspection, force control

## Abstract

This study presents an experimental robotic setup with a Stewart platform and a robot manipulator to emulate an underwater vehicle–manipulator system (UVMS). This hardware-based emulator setup consists of a KUKA IIWA14 robotic manipulator mounted on a parallel manipulator, known as Stewart Platform, and a force/torque sensor attached to the end-effector of the robotic arm interacting with a pipe. In this setup, we use realistic underwater vehicle movements either communicated to a system in real-time through 4G routers or recorded in advance in a water tank environment. In addition, we simulate both the water current impact on vehicle movement and dynamic coupling effects between the vehicle and manipulator in a Gazebo-based software simulator and transfer these to the physical robotic experimental setup. Such a complete setup is useful to study the control techniques to be applied on the underwater robotic systems in a dry lab environment and allows us to carry out fast and numerous experiments, circumventing the difficulties with performing similar experiments and data collection with actual underwater vehicles in water tanks. Exemplary controller development studies are carried out for contact management of the UVMS using the experimental setup.

## 1. Introduction

An underwater vehicle–manipulator system (UVMS) consists of an underwater robotic manipulator mounted on an underwater vehicle typically used for subsea inspection and surveillance [1,2,3]. Due to the inherent danger of manned subsea operations, the research interest in underwater robotic systems has continuously increased as UVMSs have a wide range of application areas—for instance, for object inspection, underwater welding, and valve manipulation within the offshore industry [1,2]. The underwater robot manipulators enhance capabilities of the underwater vehicles and reduce operational costs and danger to human life for the essential subsea tasks requiring physical interaction. However, designing robust controller for such a complex system is a challenge from a control point of view due to the highly dynamic coupling between the manipulator and the floating vehicle. In addition, the overall system needs to be robust against external disturbances, e.g., caused by waves or tidal streams, while the end-effector of the manipulator is interacting with the environment. These are also common problems for land-based mobile manipulators but are particularly relevant to UVMSs as the base vehicle is floating. For testing and demonstration purposes, here we consider underwater asset inspection/manipulation tasks which require maintaining a physical contact with the asset surface, where the exact location of the contact on the surface or the exact trajectory followed on the surface is not critical. This is typically the case with pipe thickness measurements, corrosion measurements, cleaning of surfaces from biological structures, and placement of (e.g., magnetically attached) sensors/devices on the surface.

In this study, it is assumed that vehicle motions, caused by environmental disturbances, are unknown for the robotic arm controller while keeping the end-effector in contact with a surface under disturbances. We have emulated environmental disturbances with realistic data that we have collected from a physical underwater vehicle floating in a water tank under the occasional impact of push movements. In addition, the physical interaction of the manipulator end-effector with the surface acts as disturbances in the motion of the vehicle. This is due to the physical coupling between the manipulator and the vehicle and transmission of the interaction force to the base through robot links. The position disturbance due to this force-impact has been computed and applied on the emulating base platform. In this way, we have obtained a realistic emulation of the underwater disturbance impacts on the robot base, by capturing the two main causes: water flow and environment interaction. As a result, a simulation environment and a physical experimental setup have been developed to interact with each other to replicate a UVMS in order to test and validate the controllers in a dry-lab environment. A force/position control method is adapted from our earlier studies [4] and an admittance based controller [5] that applies virtual dynamics at the manipulator end-effector for perpendicular force interaction with the unknown surface is implemented in this study. This admittance controller does not require knowledge of the vehicle position/velocity, the stiffness of the environment or manipulator base disturbance effects. We demonstrate the use of the setup to replicate costly underwater experiments, through an evaluation of an admittance-based controller in comparison to a PID based controller, both in simulations and in physical experiments with the hardware-based emulator.

For the problem of physical contact and surface tracing using a UVMS in the underwater environment, the authors in [6] proposed an optimized redundancy resolution scheme for operational space tracking control of the end-effector of a UVMS. In [7,8], the authors used task-priority-based redundancy resolution methods where the primary task was defined by the operational space position/velocity tracking and force tracking was proposed as a secondary objective. In [9,10], the authors proposed force/position hybrid controllers for the interaction of the end-effector of UVMS with an underwater environment. In [11], an impedance control focused on task priority redundancy solution was developed for contact force control of UVMS. However, these approaches do not consider the problems related to the disturbance effects on the underwater vehicle motion, since they always have access to the position data of the end-effector relative to an inertial base.

In [12,13,14], the problem of the physical interaction has been considered for the aerial robots, and they developed variable impedance controllers based on force estimations without using force sensors. For the general problem of hard contact interaction of robot manipulators, the authors in [15,16,17] developed dynamic adaptive hybrid impedance controllers.

In the surveys of underwater robotics [18,19], there are several simulators for the development of underwater robotics. In the TRIDENT project [20], an ROS-based open-source kinematic simulator, named UWSim, was developed for underwater robotics simulation. In [21], Gazebo was integrated into the UWSim to simulate kinematics and dynamics of underwater robots. In [22], the authors extended a Gazebo-based Unmanned Underwater Vehicle (UUV) simulator by implementing the model of hydrodynamic effects. In [10], the authors developed a hybrid simulator for underwater vehicles and manipulators with the ability to accurately simulate hydrodynamic and contact forces of the UVMS with the environment. However, these studies focused only on the development of software-based simulation frameworks to simulate the dynamics or kinematics of underwater vehicles and manipulators. However, due to the complexity of accurately modeling and simulating the physical disturbances and the interaction forces/torques with an environment, a hardware-based emulation system with physical interaction provides more realistic means of testing and validation for a UVMS. Therefore, in our study, in addition to the software-based simulation, we have a hardware emulation of underwater robotics.

Briefly, we can summarize the main contributions of this study as follows: first, we used realistic underwater vehicle movements transmitted in real time in the experimental setup or pre-recorded in a water tank environment. Next, we simulated the water current effect on floating base vehicle motion, considering both hydrodynamic and contact interaction effects. We also used a physical robotic setup with a Stewart platform and a robotic arm manipulator to emulate a UVMS. We then demonstrate the use of this system to perform fast and numerous experiments to compare control schemes for underwater asset inspection without lengthy and costly underwater experiments.

## 2. Realistic Real-Time Data Set and Transfer from Water Tank to the Land Robotic Setup

In this study, a real Falcon underwater ROV is deployed at sea in a realistic environment. This vehicle is connected through 4G to the remote lab (approximately 160 km apart) where its position and velocity (in 6 DOF) are used to drive a 6 DOF Stewart platform, see Figure 1. This setup provides a good proxy for the real experiments without the need for complex and expensive underwater hardware and integration.

As shown in Figure 1, the land robotic setup was located in the laboratory (in Edinburgh, UK) and real-time communication between the laboratory and the remote water tank (in Blyth, UK) was established through 4G routers (DrayTek Vigor 2862). During the exemplary studies, a time delay of about 0.3 s was observed. The ROV navigation data were recorded during the experiments and are reproducible on the robotic setup to evaluate future algorithm improvements.

## 3. Software-Based Simulation Platform

We have developed a UVMS simulation platform in Gazebo using an underwater vehicle and environment proposed in [22]. The simulation platform consists of a 7 DOF robot manipulator model (KUKA IIWA14) mounted on a 4 DOF underwater vehicle model (Rexrov2) and a pipe as an interaction object in the underwater environment. The force sensor attached at the end-effector of the manipulator allows us to measure the interaction force which is used to generate joint motion commands during the surface inspection. In order to move the Rexrov2 in the simulation, Gazebo uses the actual position measurements of the real Falcon ROV in the water tank. Figure 2 shows the overall underwater simulation platform; this platform was developed in Gazebo simulating a UVMS (a robot manipulator mounted on the Rexrov2 vehicle) to perform a surface inspection on a pipe. This simulation platform has been used in integration with the physical setup during the exemplary studies for controller development of contact management.

The simulator we developed is based on the UUV Simulator [22,23] consisting of Gazebo/ROS plugins with the implementation of Fossen’s equations of motion for underwater vehicles [24], 6 DOF PID controllers for ROV thrusters’ modules, ocean wave model with hydrodynamics and hydrostatic effects, and the Rexrov2 vehicle model [25]. In this way, our physical land robotic setup that will be explained next considers the impact of (simulated) water dynamics and manipulator force interaction effects on the base vehicle, along with other pre-recorded realistic position disturbances.

## 4. Physical Robotic Setup

Figure 3 shows the land robotic setup; this setup emulates a UVMS with a real KUKA IIWA14 robot manipulator fixed on the Stewart parallel manipulator platform interacting with a pipe. It is composed of a 7 DOF robot manipulator (KUKA IIWA14) to emulate an underwater robotic manipulator, a 6 DOF base vehicle (Stewart parallel manipulator) to emulate an underwater vehicle and an ATI Gamma NET FT force sensor attached to the end-effector of the manipulator for the contact management. Since pipes are one of the most common objects to be interacted within the offshore subsea environment [26,27], a PVC vent pipe with a diameter of 500 mm and a thickness of 4 mm was placed in front of the land robotic setup to emulate the underwater object that the UVMS’s end-effector is supposed to inspect. In the exemplary studies, the real Falcon ROV’s actual position data from the water tank was used to move the Stewart platform. It should be noted that the actual position measurement of the Falcon ROV was only used to move the platform and not to control the manipulator. Since the communication is unilateral and open-loop control is implemented on the Stewart platform, the communication time delay between the two locations did not impact the test and verification of control quality.

## 5. Interaction of Physical Robotic Setup-Realistic Data-Simulation Platform

Generally in a UVMS, once the end-effector of the manipulator contacts an object, the interaction forces and torques at the contact point would result in reaction forces (and torques) on a floating base vehicle that disturbs its position (and orientation) with respect to the inspected object. Therefore, in our physical robotic setup, the interaction forces at the end-effector of the (KUKA) manipulator should be accounted for and reflected to the (Stewart) base platform as a position disturbance. In the simulation platform, we simulated the position disturbance on the floating vehicle due to the real-time force interaction of the end-effector, using the model of a Rexrov2 vehicle with dynamic parameters and PID controllers on its thrusters [22,25]. Afterwards, we embedded these disturbances on top of the previously recorded water wave disturbances (realistic data set) as shown in Figure 4. While the water wave disturbances were pre-recorded, the disturbances due to interaction were dynamically changing in real-time according to the actual interaction of the manipulator in the physical robotic setup. For that purpose, first the force/torque (F/T) interaction that would occur between the underwater manipulator base and vehicle are computed using the end-effector F/T measurements, and then the resultant F/T on the center of mass of the vehicle are computed and superimposed on the force and torque resulting from the thrusters of the Rexrov2 in the simulator. The overall computed movement of the Rexrov2 in the simulator is added to the recorded realistic movement of Falcon ROV in the water tank, and the result is finally transferred to the physical Stewart platform emulating the vehicle movement in the dry-lab.

Overall, we measure the force at the tool-tip in the physical robotic setup and feed this measurement into the simulation platform. The simulator computes the movement of the base under this impact (the simulator considers the models of the robot arm [28] and the base vehicle [23] along with the water dynamics [22,24]). We then merge the simulator vehicle position with the designed disturbance effect (i: no disturbance, ii: sinusoidal movement in each direction, iii: realistic underwater disturbance recorded on an underwater vehicle; as will be explained in the following sections) and send the merged position signal in a feed-forward way to the Stewart platform in the physical robotic setup.

The closed-loop force/position controllers in the operational space are applied only to the KUKA manipulator for the contact management. On the other hand, the floating base (Stewart platform) is independently controlled by the open-loop position commands provided from real position data of the Falcon ROV due to water wave disturbance and simulated position data of the Rexrov2 due to contact interaction disturbance. All the software implementation of the real-time controllers of the robotic setups, reading of the F/T measurements of the sensor, interacting with the Gazebo simulator, and communicating with the ROV in the water tank through 4G routers was conducted in C++ under Ubuntu with the Robot Operating System (ROS) middleware running at 1 kHz. A marker was attached to the end-effector through a compliant adapter. When the end-effector tool contacts and makes a tracing movement on the pipe surface, the ATI’s Gamma F/T sensor attached between the end-effector and the tool measures the forces and torques in 3 translational directions [*x*
*y*
*z*] and three rotational directions [*α*
*β*
*γ*] in the operational space at the frequency of 1 kHz. The KUKA robot manipulator uses the KUKA Robot Controller (KRC) that operates at 1 kHz as a client on a remote workstation. The Stewart platform is connected to a real-time QNX control box running at 30 Hz which in turn connects to the central control computer.

## 6. Exemplary Studies for Development of Contact Management Controllers

The experimental setup was evaluated with the force/position hybrid control architectures of [4,5] for the contact management. The aim of the force controller is to ensure that the end-effector of the robot manipulator is in contact with the environment perpendicularly via applying a linear reference force in the *z* translational direction (a dynamically changing direction always perpendicular to the unknown surface) and a zero torque in roll (α) and pitch (β) rotational directions in the local (tool) frame. Additionally, the position controller enables the end-effector to follow the desired motion in the *x* and *y* directions in the local frame. In these hybrid control methods, the force and position controls are designed independently in dynamically changing local frame directions according to the shape of the surface to generate the end-effector velocity commands in each iteration. This approach is an adaptation of the operational space formulation proposed in [29]. The control strategy in [4] is for fixed-based robot manipulators where a standard proportional (P) controller was used to control perpendicular force interaction and surface trajectory tracking. In [5], taking into account the unknown disturbance effect of the floating base vehicle to the position of the robot manipulator, the control architecture is enhanced via an admittance control approach.

The proposed system has been evaluated in three different application scenarios where in each case the platform commanded to carry out distinct motions (*i*: no movement on the Stewart platform; *ii*: sinusoidal movement in each Cartesian direction with a position change of 0.1sin(2πTt) m in x,y,z translational and 0.1sin(2πTt) rad in α,β,γ rotational directions with *T* = 8 ms sampling period; *iii*: the actual Cartesian pose of a real ROV submerged in a water tank). In scenarios II and III, Rexrov2’s position in the simulator is also added to the movement of the Stewart platform to account for the disturbance effects of hydrodynamics and contact interaction on base vehicle movement. In all scenarios, the performance comparison between the admittance controller [5], the P force controller in [4] and the PID force controller are presented. It should be noted that, when these force controllers are separately implemented in the end-effector’s *z* translational, α and β rotational directions, simultaneously the same PD position controller is implemented to the end-effector’s *x* and *y* translational directions in all scenarios. For the admittance controller for the perpendicular force contact interaction, the general inertia and damper coefficients were chosen as 0.5 Kg and 100 Ns/m, respectively. For comparison purposes to the case of force control, the PID control gains were used as $K_{P}=0.05$, $K_{D}=0.5$, and $K_{I}=0.002$.

### 6.1. Application Scenario-I

In the first experiment, the platform is fixed in the global frame for benchmarking. The P, PID, and the admittance controllers are separately implemented on the manipulator for force control. As shown in Figure 5a,b and Figure 6a, the end-effector perfectly tracks the pre-specified trajectory as projected on the 3D surface, and Figure 7c illustrates that it continuously applies the desired force −2 N on the pipe surface.

### 6.2. Application Scenario-II

In this scenario, a pre-defined sinusoidal Cartesian position along with the Rexrov2 movement due to the hydrodynamic and contact interaction forces effecting the base is commanded to the platform. The purpose here is to observe the manipulator behavior when the vehicle is subject to a known (sinusoidal) disturbance movement (without the complicated disturbance movement of realistic underwater data and without the impact of force interaction of the manipulator). The P, PID, and the admittance controllers are separately implemented on the manipulator for force control. Then, the results of the three force control methods are compared; see Figure 6b and Figure 8. The sinusoidal movement deviates the end-effector trajectory from the intended raster movement. However, as expected, the end-effector remains in contact with the pipe staying perpendicular to the surface and applies a force in the *z* direction (Figure 9c), no matter how far it deviates from the pre-specified trajectory in the *x* and *y* directions (Figure 6b and Figure 8).

### 6.3. Application Scenario-III

In this scenario, the Stewart platform moves according to the actual Cartesian pose of the real ROV in the water tank plus the Rexrov2 movement in the Gazebo simulator due to the contact interaction disturbance. Here, as in the previous scenarios, the P, PID, and the admittance controllers are separately implemented to the floating-based manipulator system. Figure 10 and Figure 11c show the 3D actual end-effector trajectories on the pipe for the admittance controller. The movement of the Stewart platform produces a disturbance effect to the base of the KUKA manipulator, but the admittance controller still keeps the end-effector perpendicularly in contact with the pipe as shown in Figure 12c and completes the trajectory tracking within the working space of the pipe surface. However, unlike in the previous scenarios (I and II), the P and PID controllers fail to maintain continuous end-effector contact in the presence of realistic disturbances. Since the Stewart platform mimics the ROV motions through the water wave disturbance and contact interaction disturbance, the actual trajectory tracking positions of the end-effector are different from the pre-specified trajectory.

### 6.4. Discussion

Before the evaluations, the PID gains were tuned in order to get the best performance possible. The main challenge was to manage the trade-off between stability in contact and fast recovery in case of loss of contact with the surface. For instance, when the system lost contact between the end-effector and the pipe surface, a low P gain resulted in the controller taking significant time to recover the contact. On the other hand, when the robot’s end-effector was in contact with the pipe, a large P resulted in instability and frequent cycles of loss-and-recovery of the contact. Therefore, by trial-and-error, the best PID control gains that gave better results than the pure P controller were identified. While the base of the robot manipulator is constantly in motion, the end-effector of the robot manipulator with a highly sensitive force sensor is in constant interaction with an object with an unknown surface and is constantly moving in all directions. Therefore, especially during this interaction, which takes place perpendicular to the surface, the vibrations that occur, as seen in Figure 6, Figure 9 and Figure 11, are caused by the measurements of the very sensitive force sensor. As a result of the advantages of force controllers, these vibrations are minimized.

In Scenario-I, since there is no disturbance on the base movement, the continuous contact and the trajectory tracking of the end-effector is achieved as expected. The mean square force errors ${\left(f\left(z\right)-f_{d}\left(z\right)\right)}^{2}$ and the standard deviations in the *z* perpendicular direction are given in Table 1 for each experimental scenario. The case for Scenario-I constituted a reference in order to compare the impact of disturbances on the base platform. Both the P controller as proposed in [4] and the PID controller designed in this study functioned as well as the admittance controller proposed in [5] (see Figure 7 and Table 1 (I)). However, in Scenario-II, the results with the admittance controller were significantly improved in comparison with the results with the P and PID controllers, (see Figure 9 and Table 1 (II)).

In the realistic Scenario-III, the controller needs to handle the movements of the base that suddenly change in different directions during the movement of the actual ROV in the water. From Figure 12 and Table 1 (III), it is observed that the deviation from the reference value is significantly less with the admittance controller compared to the other two controllers. In this scenario, various losses of contact with the pipe were observed with all three controllers (see Figure 11). However, the total duration of the loss of contact is much less with the admittance controller (even not observable on the marker trace on the pipe in Figure 11c). It is clearly seen from Figure 11a,b that there are significant losses of contact with the P and PID controllers.

As a result, as seen in Figure 12 and Table 1, when the P, PID, and admittance controllers were compared, the admittance controller has less mean squared force error and standard deviation than the P and PID controllers in fixed-based experimental (I) and floating-based experimental scenarios (II and III). Most importantly, the disturbance effects caused by the floating real ROV and the simulated ROV under the contact interaction can be much better compensated by the admittance controller compared to the P and PID controllers.

## 7. Conclusions

This study demonstrated that force/position control approaches for the physical interaction of the UVMS with underwater structures can be developed with the experimental robotics setup in a dry laboratory environment that allows us to carry out fast and numerous trial experiments. This experimental setup consists of three sub-setups. First, we used realistic underwater vehicle movements transmitted to the system in real time or pre-recorded in a water tank environment. Second, we simulated the water current impact on the floating base vehicle movement considering both hydrodynamic and contact interaction effects. Third, we used a physical robotic setup with a Stewart platform and a robotic arm manipulator to emulate a UVMS. We have demonstrated the use of this system to conduct experiments to compare control schemes for underwater asset inspection, without lengthy and costly underwater experiments. Particularly, we have shown that an admittance control scheme performs better than conventional P and PID controllers for contact and force level management in interaction with an unknown surface.

## Figures and Tables

**Figure 1 sensors-22-05827-f001:**
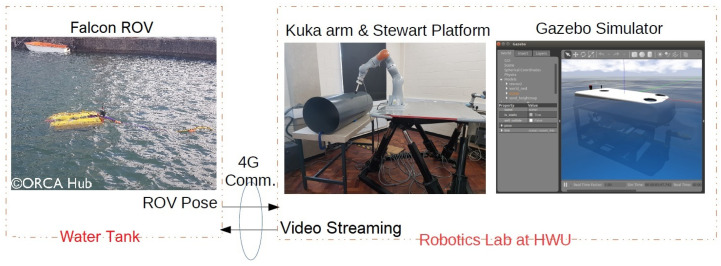
Software and hardware implementations from a real demonstration between the Robotics laboratory in Edinburgh, UK and the water tank in Blyth, UK.

**Figure 2 sensors-22-05827-f002:**
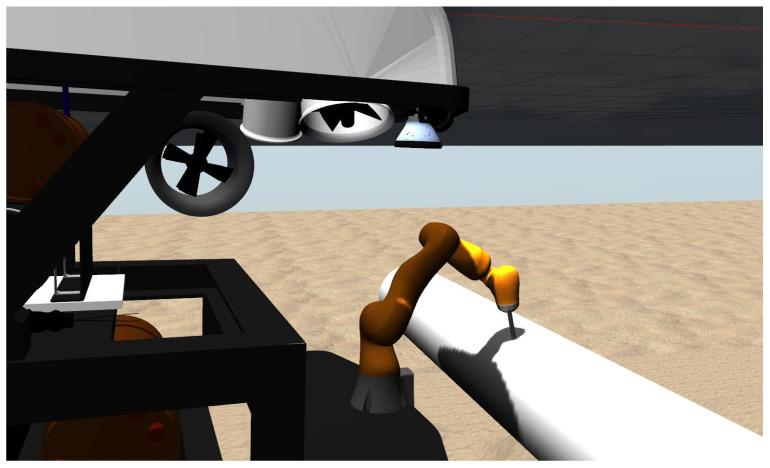
Simulating the UVMS using robotics simulation platform Gazebo. A KUKA IIWA manipulator model mounted on a Rexrov2 vehicle carries out surface inspection on a pipe.

**Figure 3 sensors-22-05827-f003:**
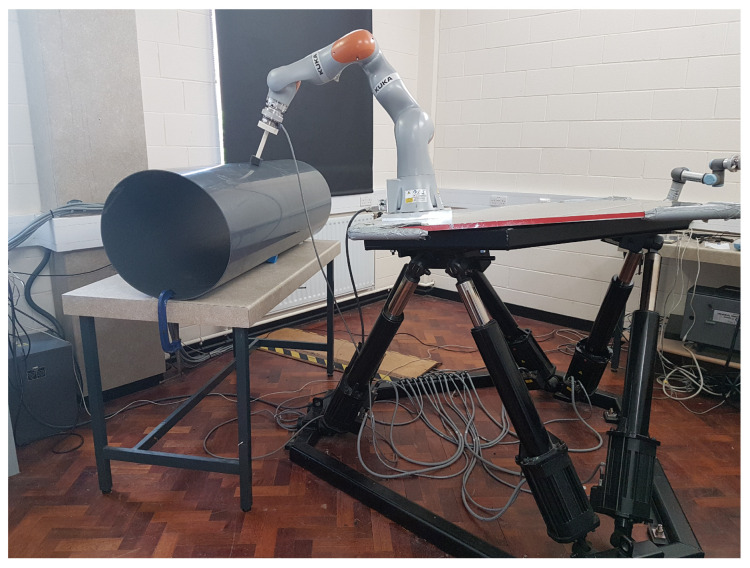
Simulating the UVMS with a real KUKA IIWA14 robot manipulator fixed on the Stewart parallel manipulator platform interacting with a pipe.

**Figure 4 sensors-22-05827-f004:**
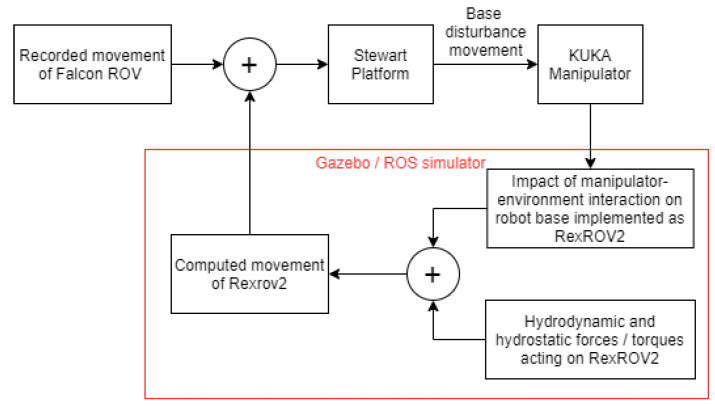
Block diagram of the floating base (Stewart platform) movement.

**Figure 5 sensors-22-05827-f005:**
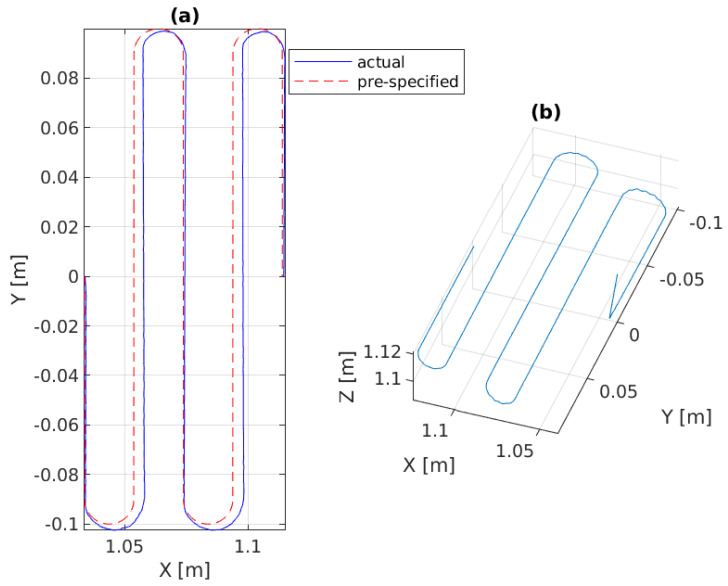
Experimental results of the admittance controller in Scenario-I: (**a**) the 2D pre-specified trajectory on XY plane versus the 2D projection of the 3D trajectory tracking, (**b**) the 3D actual end-effector trajectory on pipe.

**Figure 6 sensors-22-05827-f006:**
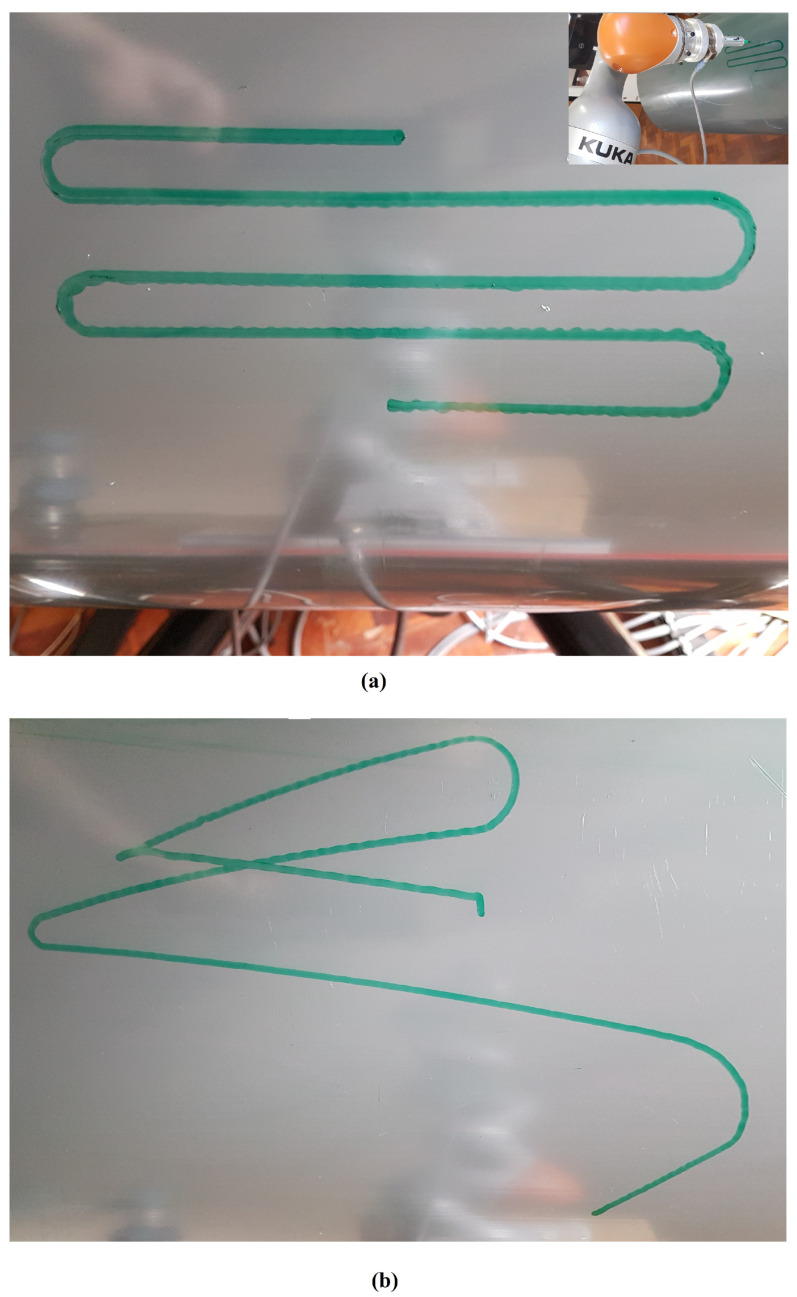
Trajectory drawing pictures on the pipe (the admittance controller was implemented): (**a**) fixed-based manipulator in Scenario-I, (**b**) floating-based manipulator in Scenario-II.

**Figure 7 sensors-22-05827-f007:**
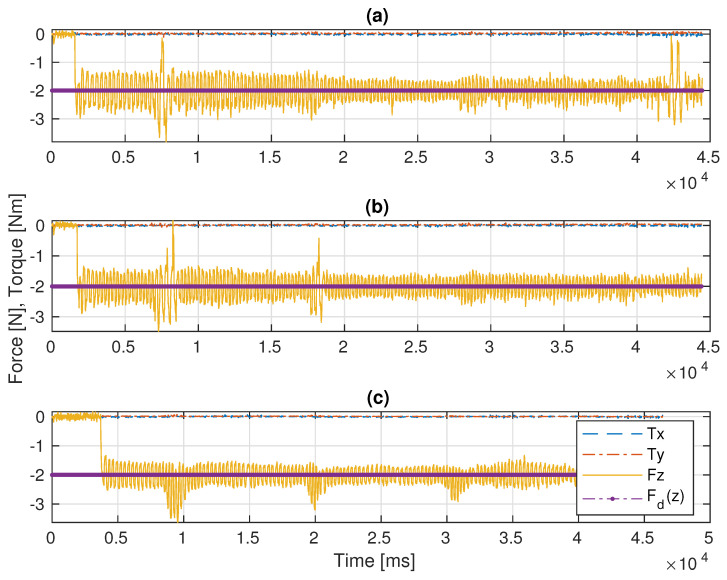
Comparative results of the F/T measurements in Scenario-I: (**a**) the P controller, (**b**) the PID controller, (**c**) the admittance controller.

**Figure 8 sensors-22-05827-f008:**
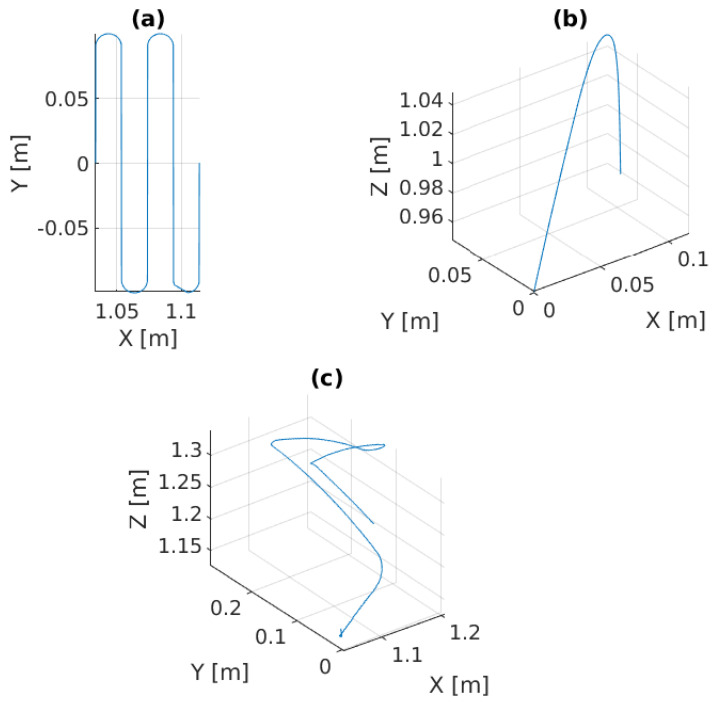
Experimental results of the admittance controller in Scenario-II: (**a**) the 2D pre-specified trajectory on XY plane, (**b**) the 3D vehicle movement as disturbance effects to the robot manipulator, (**c**) the actual end-effector trajectory on pipe with respect to the global frame.

**Figure 9 sensors-22-05827-f009:**
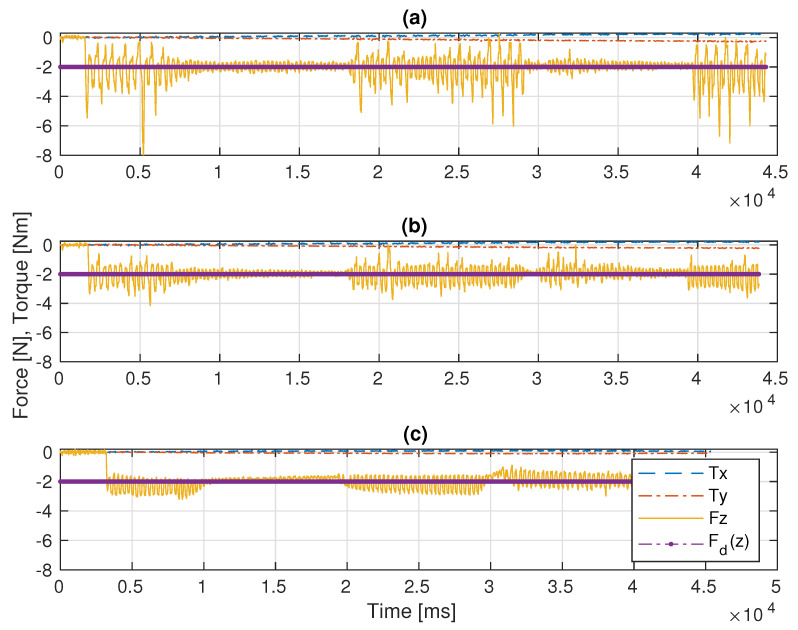
Comparative results of the F/T measurements in Scenario-II: (**a**) the P controller, (**b**) the PID controller, (**c**) the admittance controller.

**Figure 10 sensors-22-05827-f010:**
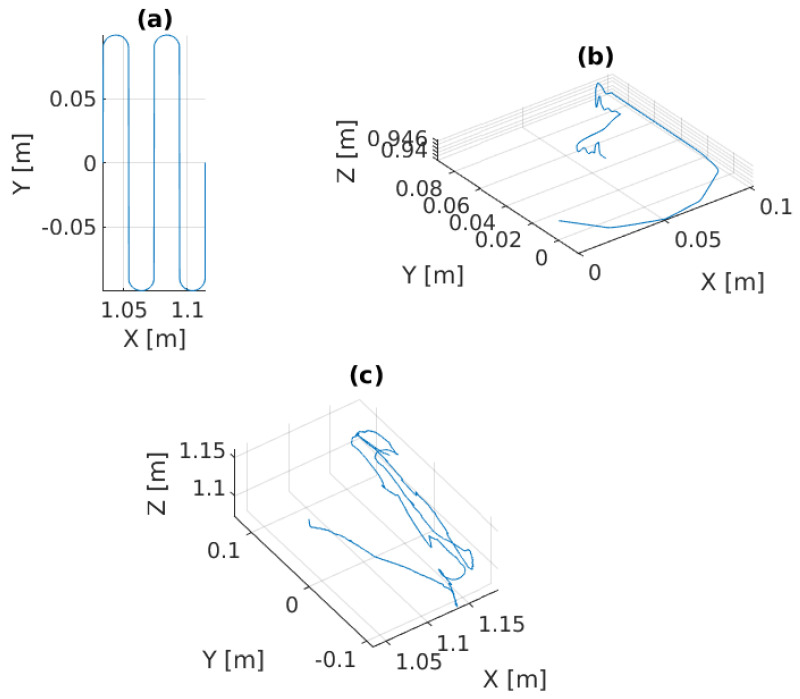
Experimental results of the admittance controller in Scenario-III: (**a**) the 2D pre-specified trajectory on the XY plane, (**b**) the 3D vehicle movement as disturbance effects to the robot manipulator, (**c**) the actual end-effector trajectory on pipe with respect to the global frame.

**Figure 11 sensors-22-05827-f011:**
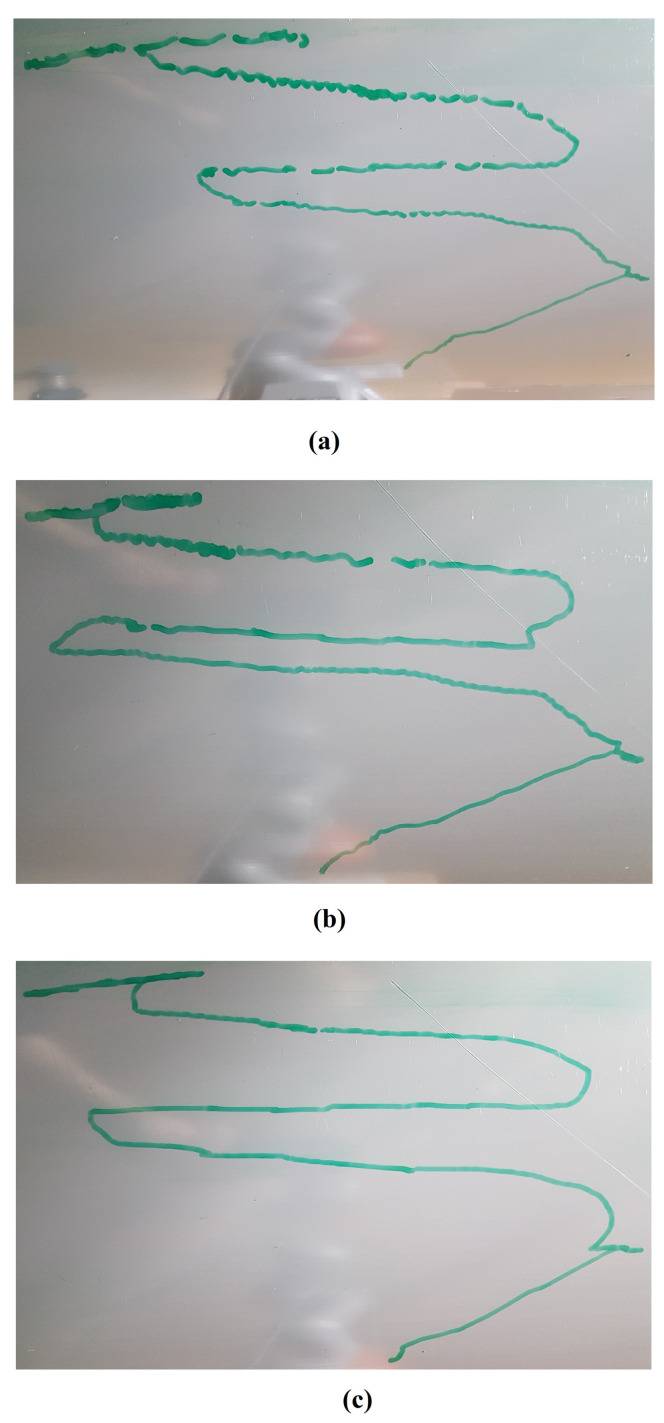
Trajectory drawing pictures on the pipe for floating-based manipulator in Scenario-III: (**a**) implementation for the P controller, (**b**) implementation for the PID controller, (**c**) implementation for the admittance controller.

**Figure 12 sensors-22-05827-f012:**
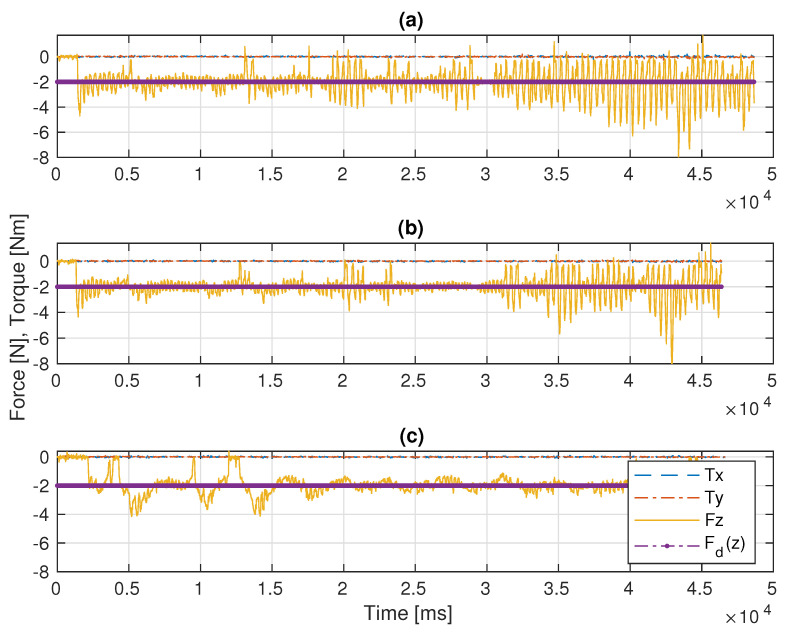
Comparative results of the F/T measurements in Scenario-III: (**a**) the P controller, (**b**) the PID controller, (**c**) the admittance controller.

**Table 1 sensors-22-05827-t001:** Mean and standard deviations of the squared force errors on the *z*-direction and the total loss of contact duration from the first contact to the end of the trajectory for the P, PID, and admittance controllers in experimental scenarios.

Application Scenarios	Force Controllers	Mean [N]	Standard Deviation [N]	Loss of Contact Duration [s]
**(I)**	P	0.19	0.29	-
	PID	0.14	0.22	-
	**Admittance**	**0.11**	**0.21**	-
**(II)**	P	0.72	2.10	0.695
	PID	0.28	0.41	0.156
	**Admittance**	**0.20**	**0.23**	**0**
**(III)**	P	1.13	2.27	4.517
	PID	0.68	2.05	2.274
	**Admittance**	**0.39**	**0.86**	**1.783**

## Data Availability

Not applicable.

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
