# Peer review of "A Robotic Experimental Setup with a Stewart Platform to Emulate Underwater Vehicle-Manipulator Systems"

_sensors, 2022, doi:10.3390/s22155827_

Round 1

Reviewer 1 Report

See attached file.

Reviewer 2 Report

The authors consider an important problem for the control of underwater vehicle manipulators with the consideration of disturbances caused by hydrodynamic forces and the interaction with the object. A major achievement of scientific research is the configuration of an original laboratory setup combining physical equipment and software tools. I believe that the article will be of interest to many researchers, because the presented results can only be obtained with special equipment, but will be useful in a number of researches in the field of underwater robotics.

Quantitative relationships in process modeling and control law are not noted in the article. However, these relationships are indicated by appropriate references in the literature.

Author Response

Thank you for these comments. Quantitative relationships in process modelling and control law are indicated in the article with appropriate references as a continuation of previous works.

The amendments to the paper are marked up using the “Track Changes” function in Latex for the convenience of the reviewers to track the changes.

Reviewer 3 Report

Please see the comments in the uploaded file.

Round 2

Reviewer 3 Report

All the comments have been considered.